🔓 | **Open Peer Review** | Human Microbiome | Research Article

# Reevaluation of the gastrointestinal methanogenic archaeome in multiple sclerosis and its association with treatment

Pei Yee Woh,[1,2] Yehao Chen,[1] Christina Kumpitsch,[3] Rokhsareh Mohammadzadeh,[3] Laura Schmidt,[3] Christine Moissl-Eichinger[3,4]

**ABSTRACT** The role of the gut archaeal microbiome (archaeome) in health and disease remains poorly understood. Methanogenic archaea have been linked to multiple sclerosis (MS), but prior studies were limited by small cohorts and inconsistent methodologies. To address this, we re-evaluated the association between methanogenic archaea and MS using metagenomic data from the International Multiple Sclerosis Microbiome Study. We analyzed gut microbiome profiles from 115 MS patients and 115 healthy household controls across Buenos Aires (27.8%), Edinburgh (33.9%), New York (10.4%), and San Francisco (27.8%). Metagenomic sequences were taxonomically classified using kraken2/bracken and a curated profiling database to detect archaea, specifically *Methanobrevibacter* species. Most MS patients were female (80/115), aged 25–72 years (median: 44.5), and 70% were undergoing treatment, including dimethyl fumarate ($n = 21$), fingolimod ($n = 20$), glatiramer acetate ($n = 14$), interferon ($n = 18$), natalizumab ($n = 6$), or ocrelizumab/rituximab ($n = 1$). We found no significant differences in overall archaeome profiles between MS patients and controls. However, treated MS patients exhibited higher abundances of *Methanobrevibacter smithii* and *M.* sp900766745 compared to untreated patients. Notably, *M.* sp900766745 abundance correlated with lower disease severity scores in treated patients. Our results suggest that gut methanogens are not directly associated with MS onset or progression but may reflect microbiome health during treatment. These findings highlight potential roles for *M. smithii* and *M.* sp900766745 in modulating treatment outcomes, warranting further investigation into their relevance to gut microbiome function and MS management.

**IMPORTANCE** Multiple sclerosis (MS) is a chronic neuroinflammatory disease affecting the central nervous system, with approximately 2.8 million people diagnosed worldwide, mainly young adults aged 20–30 years. While recent studies have focused on bacterial changes in the MS microbiome, the role of gut archaea has been less explored. Previous research suggested a potential link between methanogenic archaea and MS disease status, but these findings remained inconclusive. Our study addresses this gap by investigating the gut archaeal composition in MS patients and examining how it changes in response to treatment. By focusing on methanogens, we aim to uncover novel insights into their role in MS, potentially revealing new biomarkers or therapeutic targets. This research is crucial for enhancing our understanding of the gut microbiome's impact on MS and improving patient management.

**KEYWORDS** human microbiome, gut microbiome, archaea, archaeome, multiple sclerosis

Gut microbiome research has advanced human knowledge to understand the functional relationship between gut microbiota and a variety of chronic diseases ranging from gastrointestinal inflammation and metabolic conditions to cardiovascular and respiratory illnesses and neurological diseases, such as multiple sclerosis (MS) (1, 2).

**Peer Reviewer** Matthew E. Jennings, University of Illinois Urbana-Champaign, Urbana, Illinois, USA

Address correspondence to Christine Moissl-Eichinger, christine.moissl-eichinger@medunigraz.at.

The authors declare no conflict of interest.

See the funding table on p. 12.

MS is a chronic autoimmune disease of the central nervous system affecting 2.5 million people globally (3), and women are at a 3-fold trauma risk explained by factors, including tobacco smoking and pregnancy (4, 5). In young adults aged 20 to 40, MS causes intestinal dysfunction, cognitive impairment, and a significant loss in the quality of life (4–6). MS' commonly diagnosed subtypes include relapsing–remitting MS (RRMS), secondary-progressive MS, and primary-progressive MS. Each subtype has different symptoms of flare-ups known as relapses, exacerbations, or attacks (4), and since the year 1993, more than 20 brands of disease-modifying therapies have been approved (5). To date, the aetiology of MS is complex and probably the result of a combination of genetic (e.g., certain HLA class II haplotypes), immunological, and environmental factors (obesity, tobacco use, microbiota alterations, Epstein–Barr virus infection, or vitamin B deficiency) (4–6).

Recently, there has been increasing evidence of the gut microbiota as a regulator of immune response and brain function, suggesting that it is one of the possible environmental factors associated with the onset and development of MS (1, 7). Gut microbiota includes bacteria, archaea, eukaryotes (including fungi), and viruses with predominant bacterial phyla of Actinomycetota, Bacteroidota, Bacillota, Fusobacteriota, Pseudomonadota, and Verrucomicrobiota potentially involved in various functional interactions with and within the host (7–13). Although microbial changes in MS have been studied, many of the published studies have focused on the depletion and enrichment of intestinal bacteria and disregarded archaea, although they are important interactive components of overall microbiota functionality in the human gastrointestinal tract (14).

Archaea are unicellular microorganisms distinct from bacteria and eukaryotes due to their unique evolutionary background. Within the human gastrointestinal tract (GIT), methanogenic archaea, particularly *Methanobrevibacter* species, are the most abundant archaeal representatives (15–17). These organisms play a crucial role in microbial community dynamics by performing methanogenesis, a form of anoxic respiration that consumes bacterial fermentation by-products, such as hydrogen, carbon dioxide, formate, acetate, methanol, ethanol, and methyl compounds, to produce methane (16). This process can influence gut microbiome composition and function by alleviating hydrogen inhibition for bacterial fermentation pathways, potentially shaping microbial ecology and metabolite availability, for host and microbiome (16, 18).

The genus *Methanobrevibacter* is primarily represented by *M. smithii* and *Candidatus* M. intestini, which co-exist in the GIT. These species interact differently with bacterial partners, and their prevalence varies with host factors, such as age (15, 19, 20). However, due to their indistinguishability at the 16S rRNA gene level, many studies relying on 16S rRNA gene sequencing have struggled to resolve species-specific roles, leading to conflicting conclusions regarding their contribution to health and disease. Moreover, while *M. smithii* is relatively well-studied, *Cand*. M. intestini is only currently being described and remains largely unexplored with respect to its function in the GIT microbiome (19).

Methanogens are of particular interest due to their metabolic by-products and ecological interactions, which may influence host health. Methane, the primary end product of methanogenesis, has been associated with prolonged intestinal transit time and conditions, such as constipation. This association has been demonstrated causally in rodent models and linked to delayed stool transit observed in Parkinson's disease (21–23). Notably, constipation is a common symptom in MS, frequently leading to fecal incontinence in 39 to 73% of patients due to pelvic floor incoordination (24, 25).

Despite these findings, the involvement of methanogens in human disease remains largely speculative. Associations with conditions, such as periodontitis, Parkinson's disease, colorectal cancer, and MS, have been reported (summarized in [16]). However, methanogenic archaea cannot be not directly pathogenic, as they are, for example, lacking classic virulence factors (further details are given in [26]). Instead, their indirect effects, such as hydrogen scavenging, may create an environment that favors the growth of certain pathogenic bacteria, such as those involved in periodontitis (27).

In 2016, Jangi et al. reported a significant association of archaeal signatures from the Euryarchaeota phylum with MS disease, particularly with *Methanobrevibacter* signatures being significantly increased in overall ($n = 60$) and particularly untreated ($n = 28$) MS patients and treated MS patients ($n = 32$) showing similar levels as controls ($n = 43$) (9). On a second cohort (MS patients, $n = 41$, including MS patients untreated, $n = 10$; healthy controls, $n = 32$), indeed, an elevated breath methane content was observed in MS patients ($P < 0.018$); however, differentiation of treated and untreated patients was not provided (9).

Considering that up to 40% of the US American population are natural carriers of a higher abundance of methanogens (18, 28–30), statistical variance could also be determined by a natural variability of archaeome content. Furthermore, the lack of standardization of study protocols, small sample size, variable sequencing approaches, disease course, treatments, and genetic and environmental factors could result in large unclear situations across studies (12, 31).

In this study, we re-evaluate the association between methanogenic archaea and MS using data from a well-characterized, publicly available cohort—the International Multiple Sclerosis Microbiome Study (iMSMS) (12). Our analysis focused on a subset of the iMSMS cohort, including MS patients (untreated, $n = 35$; treated, $n = 80$) and healthy controls ($n = 115$). The iMSMS features a robust household-controlled MS-control (HHC) paired experimental design, which mitigates common biases in microbiome studies and enhances statistical power for archaeome analysis. Additionally, the use of metagenomic sequencing data in this study eliminates biases introduced by primer-based amplification, enabling an improved and unbiased classification of archaea within the gut microbiome.

## MATERIALS AND METHODS

### Study overview

We utilized the publicly available iMSMS data set (accession number PRJEB32762, provided by the iMSMS Consortium [12]) due to its large cohort size, robust household-matched experimental design, archaea-compatible DNA extraction methodology, and the use of metagenomic sequencing, which enables species-level taxonomic profiling. For this study, we selected a 1:1 ratio of MS patients and their genetically unrelated HHCs from 115 households in the iMSMS cohort. Only single sampling time points and metagenomic data were included in the analysis, with a focus on ensuring a substantial representation from a specific geographical site (Buenos Aires, Edinburgh, New York, and San Francisco; Table S1). The processing of the original fecal samples is described in detail by Zhou et al. (12).

### Sequence data processing

Metagenomic data sets were classified with kraken2 (32) (confidence threshold : 0.1) using the Unified Human Gastrointestinal Genome database (33) (UHGG, v. 2.01; available through MGnify (https://www.ebi.ac.uk/metagenomics) (34), which allows for a species-level resolution of the human archaeome (15, 35). To estimate the relative abundance of domain, family, genera, and species, we used bracken (36) (read-length: 100) (Tables S2 to S6). The outputs of kraken2/bracken were further subjected to centered log-ratio (CLR) transformation.

Please note that all taxonomic classifications provided (e.g., Methanobrevibacter_A smithii_A) follow the current Genome Taxonomy Database (GTDB) (37).

### Statistical analysis

All analyses were performed using R version 4.1.3 (38). The R packages *ggplot2* and *ggpubr* were utilized as the primary tools for data visualization and statistical testing. The figure legends also provide information regarding statistical tests and significance cutoff.

All R scripts are available in the online supplementary materials (https://github.com/YipHoChan/MS). We tested the differences in the archaea community among various clinical outcome groups by comparing the CLR-transformed relative abundance of archaea at the domain and species levels. Depending on the number of groups, either the Wilcoxon rank-sum test or the Kruskal–Wallis test with post-hoc Wilcoxon rank-sum test was chosen. On the contrary, we used the *percent_format* function from *scales* R package to transform the abundance of archaea species before plotting the compositional stacked bar charts. To compare collected host factors between different clinical outcome groups, analysis of variance (ANOVA) was performed on normally distributed continuous variables and the Kruskal–Wallis test on non-normally distributed variables. For categorical variables, we used the $\chi$ test. The correlations between *Methanobrevibacter* species and Multiple Sclerosis Severity Score (MSSS) or Expanded Disability Status Scale (EDSS) were assessed by Spearman's correlation. To assess the impact of confounding factors (fixed effects: age, sex, body mass index, BMI; random_effects: household, site), Maaslin2 (v 1.8.0) was used for not-normalized kraken2/bracken outputs.

## RESULTS

The goal of this study was to re-evaluate the association of methanogenic archaea with MS disease (9). For this, we used a public data set provided by the iMSMS Consortium (12). We selected a ratio of 1:1 MS patients and their genetically unrelated HHCs from 115 households participating in iMSMS and performed thorough, archaea-tailored profiling and statistical analyses to answer our research question.

### General characteristics of the cohort

A total of 230 fecal metagenomes from 115 MS and 115 HHC from Buenos Aires (*n* = 32 + 32, 27.8% of the included data), Edinburgh (*n* = 39 + 39, 33.9%), New York (*n* = 12 + 12, 10.4%), and San Francisco (*n* = 32 + 32, 27.8%) were analyzed (Table S1). About 70% of MS patients were female aged between 25 and 72 (median = 44.5). Among 115 MS patients, 80 (70%) received treatments with dimethyl fumarate (*n* = 21), fingolimod (*n* = 20), glatiramer acetate (*n* = 14), interferon (*n* = 18), natalizumab (*n* = 6), and orelizumab/rituximab (*n* = 1). Compared to the untreated group, the treated MS patients showed a reduced body weight [median = 68.35, interquartile range (IQR) = 58.65, 78.00] with lower MSSS (median = 2.53, IQR = 0.64, 4.26) and EDSS (median = 1.50, IQR = 0.00, 3.00), which quantifies disability in multiple sclerosis over time. In terms of disease course, 80% (92/115) of MS patients who experienced RRMS (median age = 40.5, IQR = 35.0, 49.3) were significantly younger than those with progressive MS (PMS) (median age = 55.0, IQR = 51.0, 58.5). Compared to the PMS group, these RRMS patients had significantly lower MSSS (median = 2.3, IQR = 0.66, 4.18) and EDSS (median = 1.5, IQR = 0.00, 2.62), and 77% (71/92) were treated with dimethyl fumarate (*n* = 19), fingolimod (*n* = 18), glatiramer acetate (*n* = 14), interferon (*n* = 15), and natalizumab (*n* = 5). Detailed information on the study population is presented in Table 1.

### MS prevalence, treatment status, and disease severity were not affected by high levels of methanogens

For the following analyses, the metagenomic data sets underwent taxonomic profiling following the procedures outlined in the Materials and methods section. The use of the UHGG database is well suited to capture the greatest diversity of archaeal signatures in the gut (15). Initially, we conducted a comparative assessment of archaeal abundance between MS patients and HHC. Notably, no statistically significant differences were found in the relative abundance of the archaeal community at the domain level (Fig. 1A). Both groups exhibited a predominance of *Methanobrevibacter*, alongside minor proportions of other archaeal genera, including *Methanomassiliicoccus*, *Methanosphaera*, and *Methanomethylophilus* (Fig. 1B). The species *M. smithii*, *Cand.* M. intestini (M. smithii_A, according to GTDB classification), and *Methanobrevibacter* sp900766745 (GTDB classification) collectively represented over 90% of the archaeal species in both groups

**TABLE 1** Sample characteristics for 115 MS (treated vs. untreated and PMS vs. RRMS) and 115 HHCs[c]

| | HHC | Treated | Untreated | P | MS | | |
| | | | | | PMS | RRMS | P |
|---|---|---|---|---|---|---|---|
| | 115 | 80 | 35 | | 23 | 92 | |
| **Sampling site, n (%)** | | | | | | | |
| Buenos Aires | 32 (27.8) | 27 (33.8) | 5 (14.3) | 0.038* | 3 (13.0) | 29 (31.5) | 0.171 |
| Edinburgh | 39 (33.9) | 22 (27.5) | 17 (48.6) | | 8 (34.8) | 31 (33.7) | |
| New York | 12 (10.4) | 12 (15.0) | 0 (0.0) | | 6 (26.1) | 6 (6.5) | |
| San Francisco | 32 (27.8) | 19 (23.8) | 13 (37.1) | | 6 (26.1) | 26 (28.3) | |
| **Gender, n (%)** | | | | | | | |
| Female | 42 (36.5) | 53 (66.2) | 27 (77.1) | <0.001[a] | 19 (82.6) | 61 (66.3) | <0.001** |
| Male | 73 (63.5) | 27 (33.8) | 8 (22.9) | | 4 (17.4) | 31 (33.7) | |
| **Education levels, n (%)[b]** | | | | | | | |
| High school | 28 (25.0) | 12 (15.0) | 7 (20.6) | 0.242 | 3 (13.6) | 16 (17.4) | 0.282 |
| Tertiary education | 84 (75.0) | 68 (85.0) | 27 (79.4) | | 19 (86.4) | 76 (82.6) | |
| **MS family history, n (%)[b]** | | | | | | | |
| No | 97 (86.6) | 64 (80.0) | 27 (79.4) | 0.394 | 15 (68.2) | 76 (82.6) | 0.105 |
| Yes | 15 (13.4) | 16 (20.0) | 7 (20.6) | | 7 (31.8) | 16 (17.4) | |
| **Smoke, n (%)[b]** | | | | | | | |
| Former smoker | 34 (30.4) | 28 (35.0) | 9 (26.5) | 0.459 | 10 (45.5) | 27 (29.3) | 0.645 |
| Non-smoker | 67 (59.8) | 43 (53.8) | 24 (70.6) | | 11 (50.0) | 56 (60.9) | |
| Smoker | 11 (9.8) | 9 (11.2) | 1 (2.9) | | 1 (4.5) | 9 (9.8) | |
| **Weight change, n (%)[b]** | | | | | | | |
| No | 96 (85.7) | 70 (87.5) | 27 (79.4) | 0.530 | 21 (95.5) | 76 (82.6) | 0.306 |
| Yes | 16 (14.3) | 10 (12.5) | 7 (20.6) | | 1 (4.5) | 16 (17.4) | |
| Age, years [median (IQR)] | 45.00 [37.00, 55.50] | 41.00 [35.00, 52.00] | 49.00 [40.00, 57.50] | 0.084 | 55.00 [51.00, 58.50] | 40.50 [35.00, 49.25] | <0.001[a] |
| Weight, kg [median (IQR)] | 77.00 [68.00, 88.45] | 68.35 [58.65, 78.00] | 73.00 [62.30, 81.85] | 0.001** | 69.80 [58.10, 77.55] | 69.50 [60.75, 80.00] | 0.001** |
| Height, cm [mean (standard deviation, SD)] | 173.38 (12.23) | 168.59 (11.00) | 168.22 (8.28) | 0.005* | 167.78 (9.44) | 168.65 (10.44) | 0.005* |
| BMI, kg/m$^2$ [mean (SD)] | 26.36 (4.39) | 24.82 (4.76) | 26.51 (5.11) | 0.051 | 24.30 (3.46) | 25.60 (5.19) | 0.123 |
| Disease duration, days [mean (SD)] | | 12.36 (10.92) | 10.49 (8.36) | 0.367 | 16.26 (12.48) | 10.67 (9.30) | 0.018* |
| MSSS [median (IQR)] | | 2.53 [0.64, 4.26] | 5.45 [2.64, 7.10] | <0.001** | 6.14 [5.16, 7.36] | 2.30 [0.66, 4.18] | <0.001[a] |
| EDSS [median (IQR)] | | 1.50 [0.00, 3.00] | 3.50 [1.75, 5.50] | <0.001** | 6.00 [3.50, 6.50] | 1.50 [0.00, 2.62] | <0.001[a] |
| **Treatments, n (%)** | | | | | | | |
| Control | 115 (100.0) | | | <0.001[a] | | | <0.001[a] |
| Dimethyl fumarate | | 21 (26.2) | 0 (0.0) | | 2 (8.7) | 19 (20.7) | |
| Fingolimod | | 20 (25.0) | 0 (0.0) | | 2 (8.7) | 18 (19.6) | |
| Glatiramer acetate | | 14 (17.5) | 0 (0.0) | | 0 (0.0) | 14 (15.2) | |
| Interferon | | 18 (22.5) | 0 (0.0) | | 3 (13.0) | 15 (16.3) | |
| Natalizumab | | 6 (7.5) | 0 (0.0) | | 1 (4.3) | 5 (5.4) | |

*(Continued on next page)*

TABLE 1 Sample characteristics for 115 MS (treated vs. untreated and PMS vs. RRMS) and 115 HHCs[c] (Continued)

| | HHC | Treated | Untreated | P | MS | | | |
| | | | | | PMS | RRMS | P |
| | 115 | 80 | 35 | | 23 | 92 | |
|---|---|---|---|---|---|---|---|
| Ocrevus (rituxan) | | 1 (1.2) | 0 (0.0) | | 1 (4.3) | 0 (0.0) | |
| Untreated | | 0 (0.0) | 35 (100.0) | | 14 (60.9) | 21 (22.8) | |

[a]$P < 0.001$; **$P < 0.01$; *$P < 0.05$.

[b]Sample size changes according to data availability and the reference denominators for these factors are as follows: HHC ($n = 112$), treated MS ($n = 80$), untreated MS ($n = 34$), PMS ($n = 22$), and RRMS ($n = 92$). HHC, healthy household control; MS, multiple sclerosis; PMS, progressive multiple sclerosis; RRMS, relapsing–remitting multiple sclerosis; BMI, body mass index; MSSS, Multiple Sclerosis Severity Score; EDSS, Expanded Disability Status Scale; OCD, obsessive/compulsive disorder.

[c]Comparison of subject characteristics was performed with ANOVA for normally distributed continuous variables, Kruskal–Wallis for non-normally distributed continuous variables, and χ for categorical variables. The left P values compare the HHC, treated, and untreated groups; the right P values compare the HHC, PMS, and RRMS groups (e.g., the left P value for the sampling site indicates statistical differences between the three groups, namely, HHC, treated, and untreated across the sampling sites).

(Fig. 1B). No statistically significant differences were observed between the abundances of these and other *Methanobrevibacter* species (or any other archaea) when comparing the profiles of MS patients and HHC (Fig. 1C). This conclusion was further supported by accounting for potential confounding factors (via Maaslin2, see Materials and methods), which yielded corrected $P$ values of $p_{corr} = 0.962$ for the differential abundance of the genus *Methanobrevibacter* between healthy and diseased individuals, $p_{corr} = 0.905$ for *Methanosphaera*, and $p_{corr} = 0.648$ for *Methanomassiliicoccus*. Similarly, all species-level analyses produced non-significant corrected $p_{corr}$ -values.

In the next step, we stratified subjects into high- and low-methanogen phenotypes based on the relative abundance of methanogenic archaea (e.g., *Methanobrevibacter*) at the genus and species levels (Table 2). For this purpose, we set a relative abundance of ≥0.03% as the threshold to identify the subjects with high-methanogen phenotype, as previously described (18). If the high-methanogen phenotype was predicted by the relative abundance of *Methanobrevibacter* spp. ($n = 120$) being ≥0.03%, we could not identify statistically significant associations between the prevalence of high-methanogen phenotypes and MS prevalence, treatment status, nor disease severity.

## The relative abundance of specific *Methanobrevibacter* species associated with MS treatment and severity but not disease course (RRMS or PMS)

We next investigated whether treatment of the MS patients had any effect on their archaeome. Statistically significant differences were observed between the archaeome

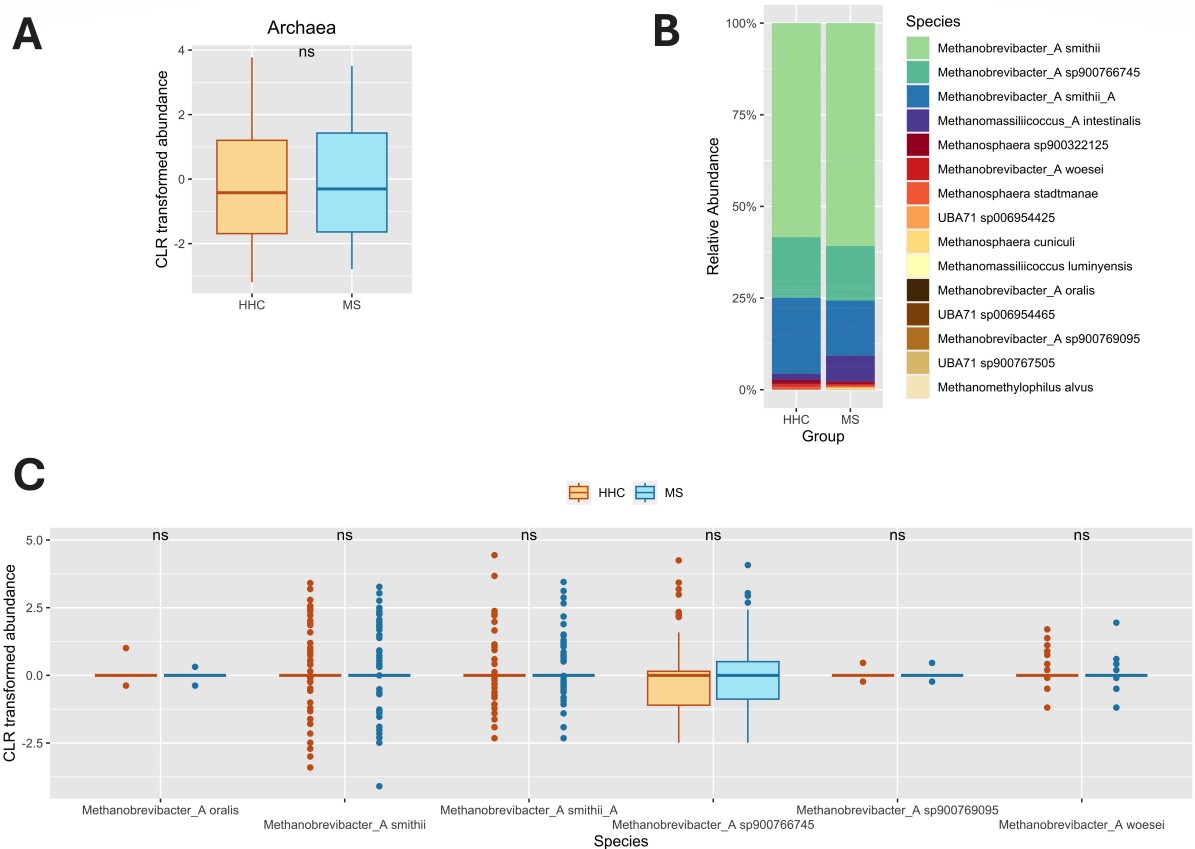

**FIG 1** Archaeal profiling of HHC and MS subjects. (A) Alterations in CLR-transformed abundance of archaea between HHC ($n = 115$) and MS ($n = 115$) at the domain level. $P$ values were determined using the Wilcoxon rank-sum test. (B) Composition of archaea abundance between HHC and MS at the species level. The visual difference of Methanomassiliicoccus_A intestinialis was not supported by a corresponding $P$ value. (C) Alterations in *Methanobrevibacter* abundance between HHC and MS at the species level. $P$ values were determined using the Wilcoxon rank-sum test. The significance was indicated as follows: ns, not significant.

of treated and untreated MS patients (Fig. 2A) mainly driven by a significantly higher abundance of *M. smithii* and *M.* sp900766745 compared to the untreated patients (Fig. 2B and C). Notably, the abundance of *Cand*. M. intestini (M. smithii_A) was not significantly different between the two MS groups.

We then aimed to examine whether the relative abundance of the archaeal community differed among subjects with PMS ($n$ = 23, Table 1) and RRMS ($n$ = 92). However, no significant differences were found between the relative abundance of *Methanobrevibacter* between the groups (Fig. 3A through C).

MS severity is determined by the score values of the EDSS and the MSSS. It shall be mentioned that the MSSS score is obtained by a normalization of the EDSS score for disease duration; thus, both scores are interdependent (33). We examined the relationship of archaea between EDSS and MSSS in all treated and untreated MS patients (Fig. 4A). We correlated the archaeal species that were significantly higher in treated MS patients with EDSS and MSSS. In treated patients, we found that *M.* sp900766745 was negatively correlated with EDSS (R = −0.27, *P* = 0.016) and MSSS (R = −0.27, *P* = 0.016) (Fig. 4). The overall results were not affected after the removal of all samples with zero relative abundances [EDSS ($R$ = −0.32, $P$ = 0.016) and MSSS ($R$ = −0.29, $P$ = 0.029)].

## DISCUSSION

For many diseases and disorders, a potential involvement of methanogenic archaea has been proposed (summarized in [16, 17]). Despite this, an increased abundance of *Methanobrevibacter* in the GIT is generally associated with positive health outcomes. Research indicates that a deficiency of methanogenic archaea is associated with several conditions, including irritable bowel syndrome with diarrhea (IBS-D) (39, 40), obesity (41), and inflammatory bowel disease (IBD) (42) (summarized in [17, 43]). Conversely, individuals with a naturally higher load of *Methanobrevibacter* (high-methanogen phenotype, up to 40% of all individuals) tend to have higher levels of formate and short-chain fatty acids (18), a lower BMI (16, 44, 45), and better overall health (17) and longevity (46). Notably, we observed a significant increase in the number of high-methanogen phenotypes in centenarians compared to younger cohorts (20), suggesting a link between increased methanogen abundance and prolonged health.

As archaea are considered non-pathogenic, they were long time deemed irrelevant to human health, shifting the focus of research to the bacterial component of the human microbiome. However, in some infectious and more local situations, like periodontitis, the abundance of *Methanobrevibacter* species correlates with disease severity (47). Elevated *Methanobrevibacter* levels have also been associated with severe constipation (e.g., constipation-type IBS, IBS-C [48]), or intestinal methanogen overgrowth (a subtype of small intestinal bacterial overgrowth, "SIBO", affecting 40% of IBS patients [49, 50]).

Given these observations, the potential link between archaea and diseases, such as MS, is of great interest and warrants further investigation. In this study, we

**TABLE 2** Correlation between high-methanogen phenotype and health condition, treatment status, and disease severity[a]

| *Methanobrevibacter* spp. | Other subjects N = 110 | High-methanogen phenotype N = 120 | P |
|---|---|---|---|
| Disease, *n* (%) | | | |
| HHC | 55 (50.0) | 60 (50.0) | 1.000 |
| MS | 55 (50.0) | 60 (50.0) | |
| Treatment status, *n* (%) | | | |
| Treated MS | 35 (63.6) | 45 (75.0) | 0.263 |
| Untreated MS | 20 (36.4) | 15 (25.0) | |
| Severity | | | |
| MSSS [median (IQR)] | 3.65 [1.01, 5.78] | 2.77 [0.64, 5.51] | 0.189 |
| EDSS [median (IQR)] | 2.00 [1.25, 4.00] | 2.00 [0.00, 3.62] | 0.195 |

[a]MSSS, Multiple Sclerosis Severity Score; EDSS, Expanded Disability Status Scale.

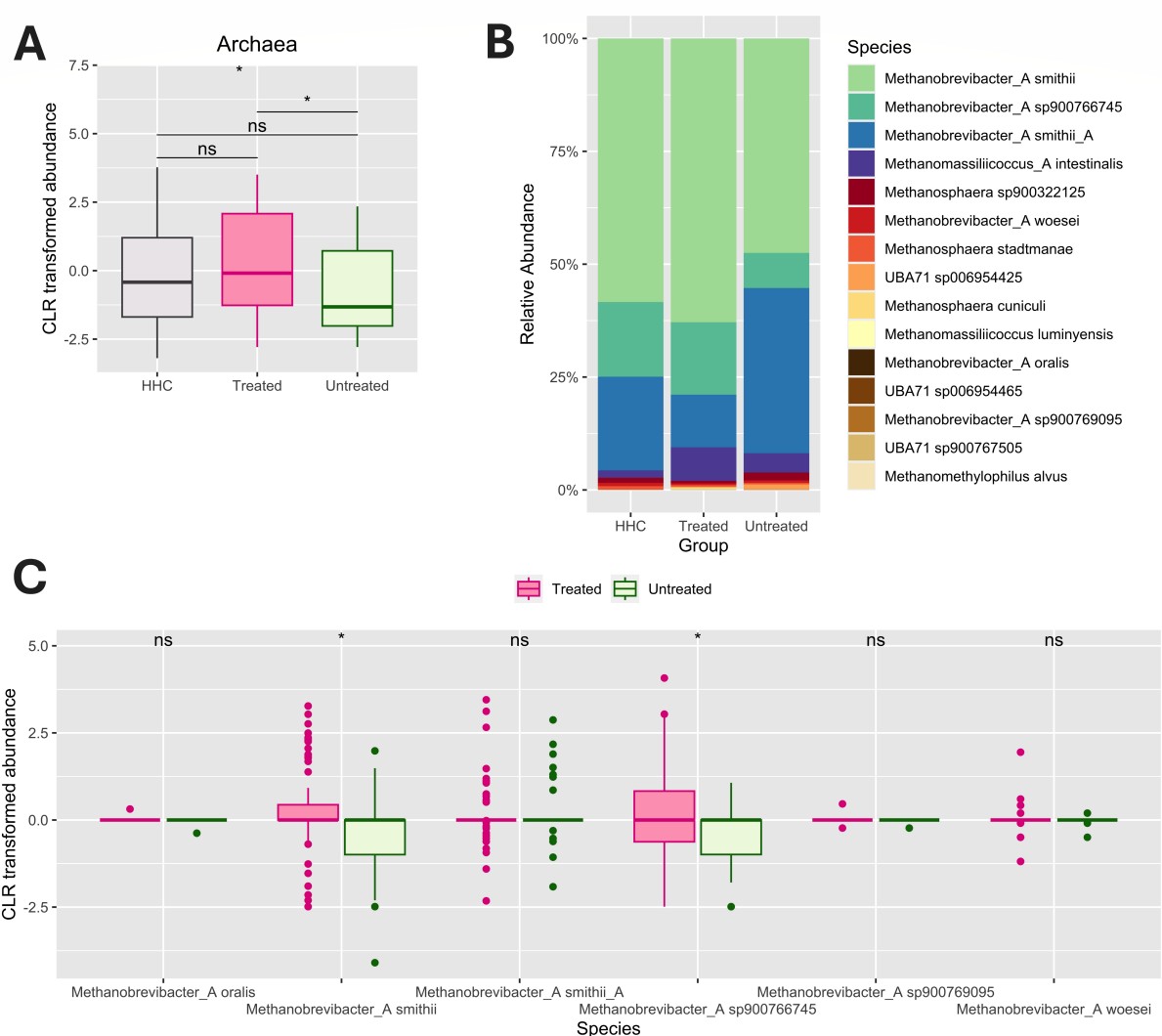

**FIG 2** Archaeal profiling of treated and untreated MS subjects. Taxonomic classification follows the GTDB (37). (A) Alterations in CLR-transformed abundance of archaea between HHC ($n = 115$), treated MS ($n = 80$), and untreated MS ($n = 35$) at the domain level. $P$ values were determined using the Kruskal–Wallis test (shown at the top) and post-hoc pairwise Wilcoxon rank-sum test (shown between boxes). (B) Composition of archaea abundance between treated and untreated MS at the species level. The visual difference of Methanomassilicoccus_A intestinialis was not supported by a corresponding $P$ value. (C) Alterations in *Methanobrevibacter* abundance between treated and untreated MS subjects at the species level. $P$ values were determined using the Wilcoxon rank-sum test. The significance was indicated as follows: *$P < 0.05$; ns, not significant.

leveraged publicly available metagenomic sequencing data and metadata from the iMSMS consortium to assess whether the abundance of archaea is implicated in MS, as previously suggested (9).

Our approach, which accounted for high-methane-emitting individuals, found no evidence, at least within the data set studied, that archaea are associated with MS. However, we did observe changes in archaeal abundance and species composition between treated versus untreated MS patients across a large, multi-center cohort of MS patients and their household controls in the United States, Europe, and South America.

Our analysis, based on public metagenomic data sets from a household-controlled experimental design with reduced confounding factors, provided substantial statistical power to detect archaeal variation and associated species across MS subtypes. Furthermore, the metagenomic approach, which focuses on the archaeal components of the microbiome, offers higher sequence coverage than 16S rRNA gene sequencing, enabling the identification of less abundant taxa and individual archaeal species in the gut.

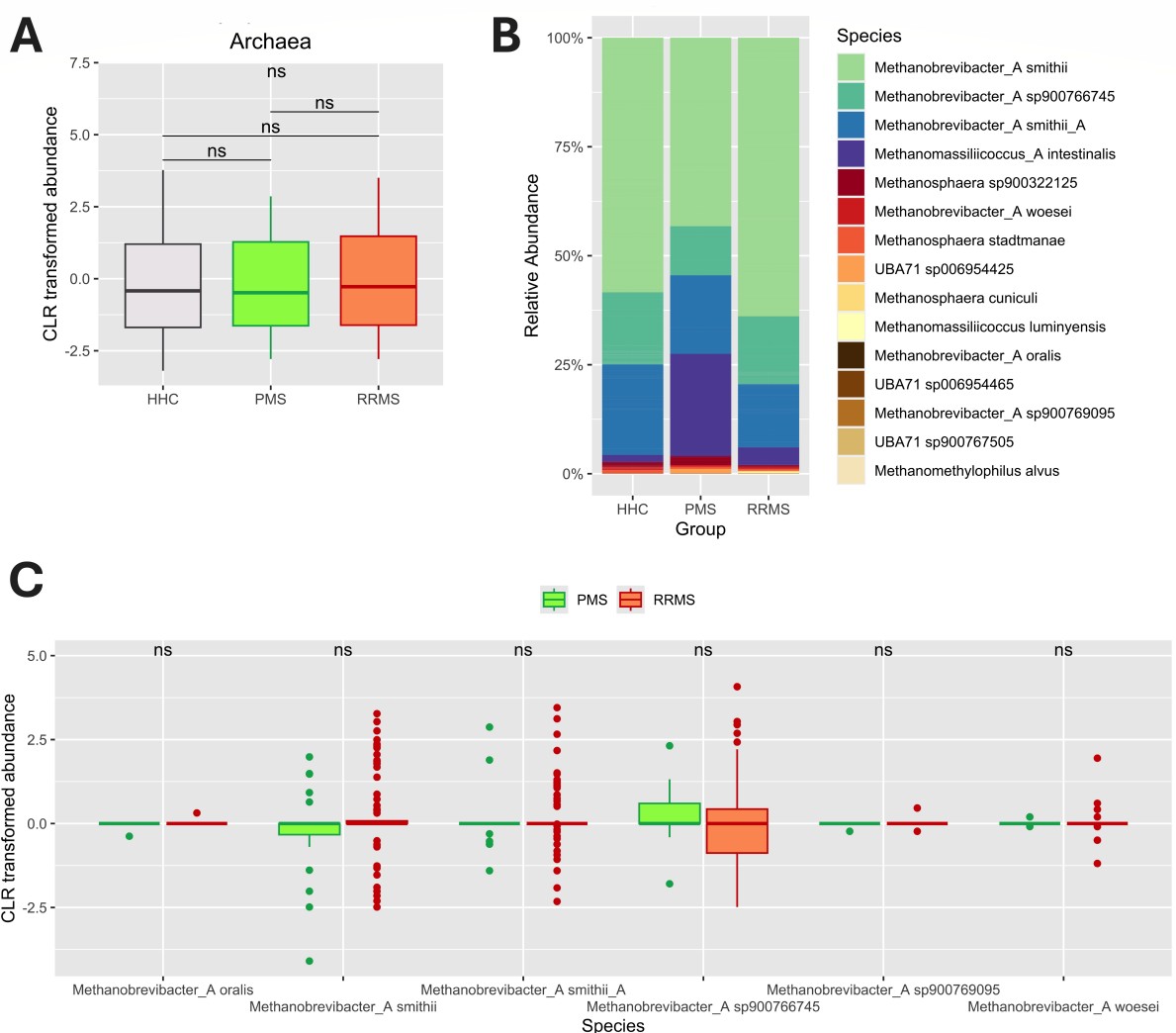

**FIG 3** Archaeal profiling of PMS and RRMS subjects. Taxonomic classification follows the GTDB (37). (A) Alterations in CLR-transformed abundance of archaea between HHC ($n = 115$), PMS ($n = 23$), and RRMS ($n = 92$) at the domain level. $P$ values were determined using the Kruskal–Wallis test (shown at the top) and post-hoc pairwise Wilcoxon rank-sum test (shown between boxes). (B) Composition of archaea abundance between PMS and RRMS at the species level. (C) Alterations in *Methanobrevibacter* abundance between PMS and RRMS subjects at the species level. $P$ values were determined using the Wilcoxon rank-sum test. The significance was indicated as follows: ns, not significant.

We noticed a significant increase in species *Methanobrevibacter smithii* and *M*. sp900766745 in treated MS patients compared to untreated MS patients, suggesting that these effects are directly related to treatment. The presence of *Methanobrevibacter* species might indicate a reconstitution of the microbiome following dysbiosis. Specifically, a higher abundance of *M*. sp900766745 was associated with lower scores of MSSS and EDSS in treated MS, suggesting a potential role of *M*. sp900766745 in ameliorating disease severity. Notably, *M*. sp900766745 lacks cultivated representatives and has not been extensively studied.

Further analysis revealed similar archaeal compositions in both PMS and RRMS patients, with no significant differences in *M*. *smithii* and *M*. sp900766745 between these subtypes.

The persistent dominance of *Methanobrevibacter* in our study suggests that alterations in gut archaeal composition in PMS and RRMS may not be exclusively associated with disease onset or progression, highlighting the need for further investigations to more accurately assess these effects.

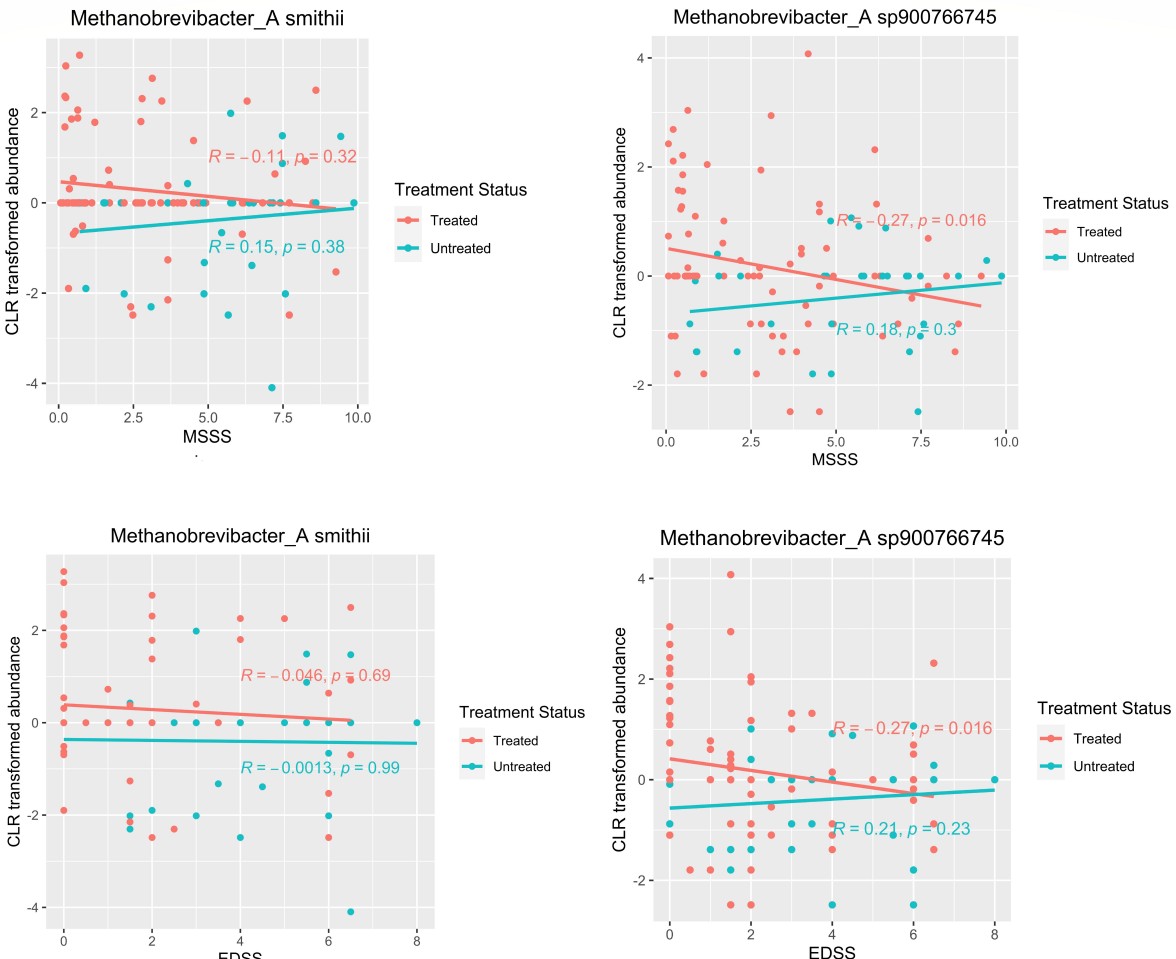

**FIG 4** Spearman's rank correlation of *Methanobrevibacter* species with EDSS and MSSS. Scatter plot showing the correlation between CLR-transformed abundance of *Methanobrevibacter* species and MSSS/EDSS by treatment status.

In summary, our findings suggest that gut archaea are likely not directly relevant to MS disease onset and progression but may serve as indicators of gut microbiome health during MS treatment. We identified two key species, *M. smithii* and *M.* sp900766745, with potential implications for MS treatment outcomes, which warrant further investigation to improve our understanding of the effects of treatment on gut archaea in MS patients.

## ACKNOWLEDGMENTS

The funders had no role in study design, data collection, and interpretation or the decision to submit the work for publication. This research was funded in whole or in part by the Austrian Science Fund (FWF) (10.55776/F83, 10.55776/P32697, and 10.55776/COE7, given to C.M.E.). For open access purposes, the author has applied a CC BY public copyright license to any author accepted manuscript version arising from this submission

P.Y.W.: methodology, formal analysis, investigation, writing: original draft, writing: review and editing, visualization, supervision, and project administration. Y.C.: methodology, formal analysis, visualization, writing: original draft, and writing: review and editing. C.K.: methodology and writing: review and editing. R.M.: methodology and writing: review and editing. L.S.: methodology and writing: review and editing. C.M.E.: conceptualization, methodology, writing: original draft, writing: review and editing, supervision, project administration, and funding acquisition.

## AUTHOR AFFILIATIONS

[1]Department of Food Science and Nutrition, The Hong Kong Polytechnic University, Hong Kong, China

[2]Research Institute for Future Food (RiFood), The Hong Kong Polytechnic University, Hong Kong, China

[3]Diagnostic and Research Institute of Hygiene, Microbiology and Environmental Medicine, Medical University of Graz, Graz, Austria

[4]BioTechMed Graz, Graz, Austria

## AUTHOR ORCIDs

Pei Yee Woh  http://orcid.org/0000-0001-5950-7883
Christina Kumpitsch  http://orcid.org/0000-0002-2077-2839
Christine Moissl-Eichinger  http://orcid.org/0000-0001-6755-6263

## FUNDING

| Funder | Grant(s) | Author(s) |
| --- | --- | --- |
| Austrian Science Fund (FWF) | F83, P32697, COE7 | Christine Moissl-Eichinger |

## AUTHOR CONTRIBUTIONS

Pei Yee Woh, Conceptualization, Data curation, Formal analysis, Funding acquisition, Investigation, Methodology, Project administration, Writing – original draft, Writing – review and editing | Yehao Chen, Data curation, Formal analysis, Methodology, Visualization, Writing – original draft, Writing – review and editing | Christina Kumpitsch, Methodology, Visualization, Writing – original draft, Writing – review and editing | Rokhsareh Mohammadzadeh, Methodology, Writing – review and editing | Laura Schmidt, Methodology | Christine Moissl-Eichinger, Conceptualization, Formal analysis, Funding acquisition, Investigation, Methodology, Project administration, Writing – original draft, Writing – review and editing

## DATA AVAILABILITY

Data and scripts that support our findings are available in our GitHub Repository https://github.com/YipHoChan/MS.

## ADDITIONAL FILES

The following material is available online.

### Supplemental Material

**Supplemental tables (Spectrum02183-24-s0001.xlsx).** Tables S1 to S6.

### Open Peer Review

**PEER REVIEW HISTORY (review-history.pdf).** An accounting of the reviewer comments and feedback.

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
