## [Reviewer comments · Microbiology Spectrum]

Microbiology Spectrum

Reevaluation of the Gastrointestinal Methanogenic Archaeome in Multiple Sclerosis and Its Association with Treatment

Pei Yee Woh, Yehao Chen, Christina Kumpitsch, Rokhsareh Mohammadzadeh, Laura Schmidt, and Christine Moissl-Eichinger

Corresponding Author(s): Christine Moissl-Eichinger, Medizinische Universität Graz

Review Timeline:

Submission Date:	September 9, 2024
Editorial Decision:	October 30, 2024
Revision Received:	January 8, 2025
Accepted:	February 2, 2025

Editor: Henning Seedorf

Reviewer(s): Disclosure of reviewer identity is with reference to reviewer comments included in decision letter(s). The following individuals involved in review of your submission have agreed to reveal their identity: Matthew E Jennings (Reviewer #1)

Transaction Report:

DOI: <https://doi.org/10.1128/spectrum.02183-24>

Re: Spectrum02183-24 (Reevaluation of the Gastrointestinal Methanogenic Archaeome in Multiple Sclerosis and Its Association with Treatment)

Dear Dr. Christine Moissl-Eichinger:

Thank you for the privilege of reviewing your work. Below you will find my comments, instructions from the Spectrum editorial office, and the reviewer comments.

Revision Guidelines

Sincerely,
Henning Seedorf
Editor
Microbiology Spectrum

Reviewer #1 (Comments for the Author):

Lines 54-69: The font for citations in the first paragraph are different from the main text, and from the font used for citations in the rest of the manuscript.

Line 74: The phyla names are out of date. Actinobacteria = Actinomycetota, Bacteroidetes = Bacteroidota, Firmicutes =

Bacillota, Fusobacteria = Fusobacteriota, Proteobacteria = Pseudomonadota, and Verrucomicrobia = Verrucomicrobiota.

Line 84: I would not describe archaea as "completely different" from the other two domains of life.

Table 1: I'm not clear on what the p values are comparing for some of the information in the chart. For example, what is the p-value indicating for sampling site? Is it comparing the treated vs untreated populations or the populations from each sampling site? A little more clarity would be helpful.

Line 306: I think the back half of this sentence was deleted.

Reviewer #2 (Comments for the Author):

The manuscript "Reevaluation of the Gastrointestinal Methanogenic Archaeome in Multiple Sclerosis and Its Association with Treatment" by Woh et al. presents the reanalysis of previously published fecal metagenomes from a Chinese cohort of 115 patients treated or untreated for multiple sclerosis (MS) compared to 115 healthy controls, with the goal to test previously suggested associations of the disease with archaea, specifically methanogens.

The most important findings from the manuscript are:

1. MS patients and healthy controls did not differ in archaeome profiles and the relative abundance of methanogens did not correlate with MS, treatment status or disease severity.
2. However, the fecal relative abundance of *M. smithii* and *M. sp900766745* was increased in treated compared to untreated MS patients.
3. And the fecal relative abundance of *M. sp900766745* was negatively correlated to MS severity.

The subject of the study, i.e. is the link of archaea/methanogens to MS, is interesting, although the authors never elaborate on the theoretical background and mechanistic explanations of this suggested link. Similarly, the findings from this reanalysis of a comparatively large dataset is important, but to what extent the underlying dataset and resulting findings are representative for other patient populations remains unclear.

Major concerns:

1. Background: The authors should provide more background on their study rationale and discuss the role of archaea and specifically methanogens for gut microbiome composition, function and health. Why do they assume relevance of this particular component for the microbiome and specifically for MS? Without this background, the study focus is not clear and seems somewhat arbitrary.
2. Methods: The authors should provide more background, details, and justification for the used dataset and applied method for methanogen/archaeal detection. The method (Kraken, UHGG) should be cited in the Results (ll. 159). How reliable are Kraken or the "Unified Human Gastrointestinal Genome database" (UHGG) for the identification and classification of archaea or methanogens? How do they compare to other methods (e.g. Metaphlan)? Are there other factors that could have influenced the results (e.g. DNA isolation)? How comparable is this dataset for other fecal (MS) metagenomes? Are the bacterial microbiome profiles similar to other studies and datasets? - Could the reported findings be biased because of the specific dataset and method?
3. Confounding factors: Table 1 shows other patient parameters that significantly differ between treated and untreated patients and MS types (PMS vs. RRMS), such as weight, height, or age. The authors should try to control for these factors in their analysis or at least discuss more prominently them as potential confounders and alternative explanations for their findings.
4. Discussion, l. 315 "However, we did observe changes in archaeal abundance and species composition [...] between relapsing-remitting MS (RRMS) and progressive MS (PMS)" - That seems wrong. Text and Fig. 3 state that "no significant differences were found between the relative abundance of *Methanobrevibacter* between the groups"

Minor concerns:

Abstract details: The authors should provide more background and method details in the abstract. It should be mentioned that the underlying data are metagenomics. How were archaea/methanogens detected? What is the suggested relevance of archaea/methanogens for microbiome function or health or those of MS patients?

Introduction

l. 59: Explain "women are at a 3-fold trauma risk".

l. 94: use italics for *M. intestini*

Results

I. 173: "experienced significant weight loss" - rephrase, as the study is cross-sectional. The authors have no information about weight changes.

Table 1: Specify statistical test and comparison in the table legend. Is the first p-value for the comparison of treated vs. untreated or amongst the three groups of healthy, treated and untreated?

Fig. 3B: It seems like *Methanomassiliicoccus intestinalis* (in violet) shows a huge difference between the groups but this is not mentioned or discussed in the text. Please elaborate.

Fig. 4: The correlations include large numbers of samples with what appear to be zero values. Do those affect the correlations? Please also show R and p-values for all remaining samples without samples have zero relative abundances.

Reviewer #3 (Comments for the Author):

The role of archaeome in human health and disease remains elusive, but it is likely to be significant as the archaeome and methanogens in particular have been associated with several diseases. In this work, the authors set out to establish their association with another disease - multiple sclerosis (MS). This is done by reanalyzing the metagenomic dataset from the iMSMS study, with a focus on changes in relative abundance of archaea and methanogens 1) between controls and MS patients (total and subtypes), 2) between treated and untreated MS patients, and 3) across severity levels of treated and untreated MS. The authors concluded that A) at the domain level, no significant difference between controls and MS patients regardless of subtypes, but the difference was significant between treated and untreated MS patients; B) at the species level, *M. smithii* and *M. sp900766745* were significantly different between treated and untreated MS patients; and C) *M. sp900766745* was further correlated with MS severity in treated patients. The results could be useful for developing biomarkers for MS treatment and stimulate new archaeal research in the gut-brain axis. Overall, this is a straightforward and concisely presented study. However, the work can benefit from additional analysis and interpretation of the data.

Major Concerns

- 1) In the original iMSMS study, 500+ MS patients were sampled. However, the current study only chose a subset of these patients without providing a reason.
- 2) Table 1, several traits have no units listed such as age, weight, height, disease duration, etc.
- 3) Figure 1B seems to show a substantial difference in the relative abundance of *Methanomassiliicoccus_A intestinalis* (purple) between controls and MS patients. This can also be seen in figure 2B between the healthy controls and both the treated and untreated MS patients, as well as in figure 3B between PMS and RRMS. However, the authors did not address this obvious observation at all. It is likely that the archaeome composition (PCA analysis) could be different in some cases. If so, a detailed statistical analysis focusing on *M. intestinalis* as well as other less obvious methanogens should be conducted.
- 4) The authors really need to educate the audience about the current state of *Methanobrevibacter* classifications. This reviewer understands that there have been new developments in the genome-based taxonomy, but simply adding '_A' in multiple species names is very confusing and counterproductive. There are also several code-only names (e.g., UBA71), what are they anyway?
- 5) Statistical significance was found here and there, but there was a lack of discussion on biology to postulate the whys.

Minor Concerns

Line 44 add word "diagnosed" before "worldwide".

Line 56 change "inflammatory" to "inflammation".

Line 65 add "have been approved" to the end of the sentence.

Line 99 change "casually" to "causally".

Line 307 add comma after "observations"

Line 200 - add quotation marks to *Candidatus*, and *intestini* should not be italicized.

Line 218 - use correct citation.

The resolution of the figures is not high.

Fig. 1B - Y-axis texts were cut off.

Figure 2A - there is a significance asterisk floating in the top of the graph - what does it represent?

ASM Spectrum

REVIEWER'S CHECKLIST

Reevaluation of the Gastrointestinal Methanogenic Archaeome in Multiple Sclerosis and Its Association with Treatment

by

Pei Yee Woh, Yehao Chen, Christina Kumpitsch, Rokhsareh Mohammadzadeh, Laura Schmidt, Christine Moissl-Eichinger

WRITTEN COMMENTARY

Abstract and Introduction

The Abstract accurately describes the contents of the paper, and I have no recommendations for it. The Introduction gives an overview of the importance of multiple sclerosis, and the various subtypes. Then an overview of the importance of the gut microbiome, with emphasis on archaeal species. The authors then outline why the data analysis was done and why they chose the dataset they did.

Lines 54-69: The font for citations in the first paragraph are different from the main text, and from the font used for citations in the rest of the manuscript.

Line 74: The phyla names are out of date. Actinobacteria = Actinomycetota, Bacteroidetes = Bacteroidota, Firmicutes = Bacillota, Fusobacteria = Fusobacteriota, Proteobacteria = Pseudomonadota, and Verrucomicrobia = Verrucomicrobiota.

Line 84: I would not describe archaea as “completely different” from the other two domains of life.

Materials and Methods

I had no issues with this section.

Results

Table 1: I'm not clear on what the p values are comparing for some of the information in the chart. For example, what is the p-value indicating for sampling site? Is it comparing the treated vs untreated populations or the populations from each sampling site? A little more clarity would be helpful.

Discussion

Line 306: I think the back half of this sentence was deleted.

Response to reviewers

We are grateful for all comments and suggestions. Please find below our responses to your questions and proposed corrections.

Reviewer #1 (Comments for the Author):

Lines 54-69: The font for citations in the first paragraph are different from the main text, and from the font used for citations in the rest of the manuscript.

The fonts for citations in the first paragraph and the rest of the manuscript have now been adjusted.

Line 74: The phyla names are out of date. Actinobacteria = Actinomycetota, Bacteroidetes = Bacteroidota, Firmicutes = Bacillota, Fusobacteria = Fusobacteriota, Proteobacteria = Pseudomonadota, and Verrucomicrobia = Verrucomicrobiota.

Thank you, very important point: The phyla names are now updated.

Line 84: I would not describe archaea as "completely different" from the other two domains of life.

The word "completed" has been deleted. Please note that parts of the introduction were re-written following the request of Reviewer 2 (Lines 78-129).

Table 1: I'm not clear on what the p values are comparing for some of the information in the chart. For example, what is the p-value indicating for sampling site? Is it comparing the treated vs untreated populations or the populations from each sampling site? A little more clarity would be helpful.

The first p-values compare the healthy, treated, and untreated groups. Second p-values compare the healthy, PMS, and RRMS. The p-value for the sampling site indicates there are statistical differences between the three groups across the sampling sites.

This information is now included in the Table description (Line 194, ff): *Sample characteristics for 115 MS (treated vs untreated and PMS vs RRMS) and 115 HHCs. Comparison of subject characteristics was performed with ANOVA for normally distributed continuous variables, Kruskal-Wallis for non-normally distributed variables, and Chi-square for categorical variables. The left p-values compare the HHC, treated, and untreated groups, the right p-value compare the HHC, PMS, and RRMS groups. E.g. the left p-value for the sampling site indicates statistical differences between the three groups HHC, treated and untreated across the sampling sites.*

Line 306: I think the back half of this sentence was deleted.

The sentence has been corrected.

Reviewer #2 (Comments for the Author):

Major concerns:

1. Background: The authors should provide more background on their study rationale and discuss the role of archaea and specifically methanogens for gut microbiome composition, function and health. Why do they assume relevance of this particular component for the microbiome and specifically for MS? Without this background, the study focus is not clear and seems somewhat arbitrary.

The introduction was now revised to support the importance of this study. We hope the study focus is now clearer (Lines 78-129)

2. Methods: The authors should provide more background, details, and justification for the used dataset and applied method for methanogen/archaeal detection. The method (Kraken, UHGG) should be cited in the Results (ll. 159). How reliable are Kraken or the "Unified Human Gastrointestinal Genome database" (UHGG) for the identification and classification of archaea or methanogens? How do they compare to other methods (e.g. Metaphlan)? Are there other factors that could have influenced the results (e.g. DNA isolation)? How comparable is this dataset for other fecal (MS) metagenomes? Are the bacterial microbiome profiles similar to other studies and datasets? - Could the reported findings be biased because of the specific dataset and method?

We appreciate the reviewer's comments and have expanded our manuscript to provide additional background and justification for the dataset and method used for methanogen and archaeal detection.

- The iMSMS dataset was chosen for its large cohort size, robust household-matched experimental design, compatible DNA extraction method and use of metagenomic sequencing, which provides higher resolution compared to 16S rRNA sequencing. This design minimizes environmental and technical biases while increasing statistical power. This is now more specifically explained in lines 132 to 140:

We utilized the publicly available iMSMS dataset (accession number PRJEB32762, provided by the iMSMS Consortium (12)) due to its large cohort size, robust household-matched experimental design, archaea-compatible DNA extraction methodology, and the use of metagenomic sequencing, which enables species-level taxonomic profiling. For this study, we selected a 1:1 ratio of MS patients and their genetically unrelated household controls (HHCs) from 115 households in the iMSMS cohort. Only single sampling time points and metagenomic data were included in the analysis, with a focus on ensuring substantial representation from a specific geographical site (Buenos Aires, Edinburgh, New York, and San Francisco; Supplementary Table 1).

- For taxonomic profiling, we utilized kraken2 with the UHGG database, which is specifically curated for the human gut microbiome and includes more than 1000 archaeal genomes from human gut, offering a comprehensive representation of gut archaea, including methanogens. Kraken2 is a well-validated, k-mer-based tool, and

the combination with UHGG enables sensitive and accurate classification of archaeal taxa. While alternative methods such as MetaPhlAn are commonly used, the resolution of methanogenic archaea was insufficient in our test runs performed with MetaPhlAn3 and led to severe misclassifications. Details are now given in Materials and Methods as follows (Lines 142-150):

Metagenomic datasets were classified with kraken2 (32) (confidence threshold : 0.1), using the Unified Human Gastrointestinal Genome database (33) (UHGG, v. 2.01; available through MGnify (<https://www.ebi.ac.uk/metagenomics>) (34), which allows for a species-level resolution of the human archaeome (15,35). To estimate the relative abundance of domain, family, genera, and species, we used bracken (36) (read-length: 100) (Supplementary Tables 2-6). The outputs of kraken2/bracken were further subjected to centered log-ratio (CLR) transformation.

Please note, that all taxonomic classification provided (e.g. Methanobrevibacter_A smithii_A) follow the current Genome Taxonomy Database GTDB (37).

- We further compared the archaeal profiles obtained in other datasets with amplicon sequencing/ metagenomic sequencing, and observed a high congruence with our kraken/bracken outputs, in particular when a confidence threshold of ≥ 0.1 had been applied, as done in this study.
- So, even if not shown in detail, as this goes beyond this study, we have carefully addressed the methodological approach and performed necessary comparisons. Please note, that this procedure was used in a number of our publications, including e.g. Duller et al (<https://www.nature.com/articles/s41467-024-52037-7>), or Chibani et al., (<https://www.nature.com/articles/s41564-021-01020-9>).
- Biases can always be an issue, but based on the current availability of data and techniques, we are very confident that our approach is the best approach we could take to analyze the archaeal fraction of the MS microbiome.

3. Confounding factors: Table 1 shows other patient parameters that significantly differ between treated and untreated patients and MS types (PMS vs. RRMS), such as weight, height, or age. The authors should try to control for these factors in their analysis or at least discuss more prominently them as potential confounders and alternative explanations for their findings.

We have run Maaslin2 to exclude the effect of confounding factors on our archaeome analyses. The results of Maaslin2 have supported our results, and this information is now included in the manuscript (Lines 217-222):

This conclusion was further supported by accounting for potential confounding factors (via Maaslin2, see Materials and Methods), which yielded corrected p -values of $p_{corr}=0.962$ for the differential abundance of the genus Methanobrevibacter between healthy and diseased individuals, $p_{corr}=0.905$ for Methanosphaera, and $p_{corr}=0.648$ for Methanomassiliicoccus. Similarly, all species-level analyses produced non-significant corrected p_{corr} -values.

4. Discussion, l. 315 "However, we did observe changes in archaeal abundance and species composition [...] between relapsing-remitting MS (RRMS) and progressive MS (PMS)" - That seems wrong. Text and Fig. 3 state that "no significant differences were found between the relative abundance of Methanobrevibacter between the groups"

Thank you for the comment. The sentence has been corrected.

Minor concerns:

Abstract details: The authors should provide more background and method details in the abstract. It should be mentioned that the underlying data are metagenomics. How were archaea/methanogens detected?

What is the suggested relevance of archaea/methanogens for microbiome function or health or those of MS patients?

The information on metagenomics and profiling method is now included. A statement on the relevance was added to the abstract (Lines 22-41).

The role of the gut archaeal microbiome (archaeome) in health and disease remains poorly understood. Methanogenic archaea have been implicated in multiple sclerosis (MS), but prior studies were limited by small cohorts and inconsistent methodologies. To address this, we re-evaluated the association between methanogenic archaea and MS using metagenomic data from the International Multiple Sclerosis Microbiome Study (iMSMS). We analyzed gut microbiome profiles from 115 MS patients and 115 healthy household controls across Buenos Aires (27.8%), Edinburgh (33.9%), New York (10.4%), and San Francisco (27.8%). Metagenomic sequences were taxonomically classified using kraken2/bracken and a curated profiling database to detect archaea, specifically Methanobrevibacter species.

Most MS patients were female (80/115), aged 25–72 years (median: 44.5), and 70% were undergoing treatment, including dimethyl fumarate (n=21), fingolimod (n=20), glatiramer acetate (n=14), interferon (n=18), natalizumab (n=6), or ocrelizumab/rituximab (n=1). We found no significant differences in overall archaeome profiles between MS patients and controls. However, treated MS patients exhibited higher abundances of Methanobrevibacter smithii and M. sp900766745 compared to untreated patients. Notably, M. sp900766745 abundance correlated with lower disease severity scores in treated patients.

Our results suggest that gut methanogens are not directly associated with MS onset or progression but may reflect microbiome health during treatment. These findings highlight potential roles for M. smithii and M. sp900766745 in modulating treatment outcomes, warranting further investigation into their relevance to gut microbiome function and MS management.

Introduction

l. 59: Explain "women are at a 3-fold trauma risk".

"women are at a 3-fold trauma risk" has been explained. (Line 60): MS is a chronic autoimmune disease of the central nervous system affecting 2.5 million people globally (3), and women are at a 3-fold trauma risk explained by factors including tobacco smoking and pregnancy (4,5).

l. 94: use italics for *M. intestini*

M. intestini cannot be written in italics at this stage, as it is not validly described yet. The correct writing is *Cand. M. intestini*.

Results

l. 173: "experienced significant weight loss" - rephrase, as the study is cross-sectional. The authors have no information about weight changes.

"experienced significant weight loss" has been rephrased (Line 184): *the treated MS patients showed a reduced body weight*

Table 1: Specify statistical test and comparison in the table legend. Is the first p-value for the comparison of treated vs. untreated or amongst the three groups of healthy, treated and untreated?

Statistical tests and comparison have been specified in the table legend. The first p-value for the comparison amongst the three groups of healthy, treated and untreated. This is now specified in the Table legend.

Fig. 3B: It seems like *Methanomassiliicoccus intestinalis* (in violet) shows a huge difference between the groups but this is not mentioned or discussed in the text. Please elaborate.

Thank you for the comment. Before journal submission, we did a statistical analysis for *Methanomassiliicoccus intestinalis* (in violet) between PMS and RRMS. Although *Methanomassiliicoccus intestinalis* (in violet) seems to show a huge difference between groups, no statistically significant difference was calculated. Therefore, we focused on the *Methanobrevibacter* community in our results and discussion.

We included a statement to the figure legends: *The visual difference of Methanomassiliicoccus_A intestinalis was not supported by a corresponding p-value.*

Fig. 4: The correlations include large numbers of samples with what appear to be zero values. Do those affect the correlations? Please also show R and p-values for all remaining samples without samples have zero relative abundances.

The R and p-values for all samples without zero relative abundances are shown below. Similar to the findings, in treated patients, we found that *M. sp900766745* was negatively correlated with EDSS (R=-0.32, p=0.016) and MSSS (R=-0.29, p=0.029). No significant correlation was observed in other species. This information is now included in the manuscript (Line 297):

The overall results were not affected after the removal of all samples with zero relative abundances (EDSS (R=-0.32, p=0.016) and MSSS (R=-0.29, p=0.029)).

Figure showing R and p-values for all samples without zero relative abundances.

Reviewer #3 (Comments for the Author):

The role of archaeome in human health and disease remains elusive, but it is likely to be significant as the archaeome and methanogens in particular have been associated with several diseases. In this work, the authors set out to establish their association with another disease - multiple sclerosis (MS). This is done by reanalyzing the metagenomic dataset from the iMSMS study, with a focus on changes in relative abundance of archaea and methanogens 1) between controls and MS patients (total and subtypes), 2) between treated and untreated MS patients, and 3) across severity levels of treated and untreated MS. The authors concluded that A) at the domain level, no significant difference between controls and MS patients regardless of subtypes, but the difference was significant between treated and untreated MS patients; B) at the species level, *M. smithii* and *M. sp900766745* were significantly different between treated and untreated MS patients; and C) *M. sp900766745* was further correlated with MS severity in treated patients. The results could be useful for developing biomarkers for MS treatment and stimulate new archaeal research in the gut-brain axis. Overall, this is a straightforward and concisely presented study. However, the work can benefit from additional analysis and interpretation of the data.

Major Concerns

1) In the original iMSMS study, 500+ MS patients were sampled. However, the current study only chose a subset of these patients without providing a reason.

We now explain the reasons for using a subset of the sampled MS patients (Lines 132-140):

We utilized the publicly available iMSMS dataset (accession number PRJEB32762, provided by the iMSMS Consortium (12)) due to its large cohort size, robust household-matched experimental design, archaea-compatible DNA extraction methodology, and the use of metagenomic sequencing, which enables species-level taxonomic profiling. For this study, we selected a 1:1 ratio of MS patients and their genetically unrelated household controls (HHCs) from 115 households in the iMSMS cohort. Only single sampling time points and metagenomic data were included in the analysis, with a focus on ensuring substantial representation from a specific geographical site (Buenos Aires, Edinburgh, New York, and San Francisco; Supplementary Table 1).

Please note, that we included all possible patient and control samples. Many of the data from the iMSMS dataset are based on 16S rRNA gene sequencing. Further, quite numerous patients were longitudinally sampled; these longitudinal samples were excluded due to potential problematic effect on the statistics. We further excluded the dataset from the Boston site, as only 24 samples were available.

2) Table 1, several traits have no units listed such as age, weight, height, disease duration, etc.

Units have been added to age (years), weight (kg), height (cm), BMI (kg/m²), disease duration (days), and treatment (n %).

3) Figure 1B seems to show a substantial difference in the relative abundance of *Methanomassiliicoccus_A intestinalis* (purple) between controls and MS patients. This can also be seen in figure 2B between the healthy controls and both the treated and untreated MS patients, as well as in figure 3B between PMS and RRMS. However, the authors did not address this obvious observation at all. It is likely that the archaeome composition (PCA analysis) could be different in some cases. If so, a detailed statistical analysis focusing on *M. intestinalis* as well as other less obvious methanogens should be conducted.

Thank you for the comment. Before journal submission, we did a statistical analysis for *Methanomassiliicoccus intestinalis* (in violet) between PMS and RRMS. Although *Methanomassiliicoccus intestinalis* (in violet) seems to show a huge difference between groups, no statistically significant difference was calculated. Therefore, we focused on the *Methanobrevibacter* community in our results and discussion.

We included a statement to the figure legends: *The visual difference of Methanomassiliicoccus_A intestinalis was not supported by a corresponding p-value.*

4) The authors really need to educate the audience about the current state of *Methanobrevibacter* classifications. This reviewer understands that there have been new developments in the genome-based taxonomy, but simply adding '_A' in multiple species names is very confusing and counterproductive. There are also several code-only names (e.g., UBA71), what are they anyway?

Thank you for this comment; we understand that this taxonomic classification might be confusing, but we strictly follow the classification provided by the current GTDB database. We added this information to Materials and Methods (Line 149): *Please note, that all taxonomic classification provided (e.g. Methanobrevibacter_A smithii_A) follow the current Genome Taxonomy Database GTDB (39).*

UBA71 is a representative of the *Methanomassiliicoccales*.

5) Statistical significance was found here and there, but there was a lack of discussion on biology to postulate the whys.

We agree. Our study was specifically designed to re-evaluate the potential involvement of methanogenic archaea in MS. Since our findings did not support this hypothesis, and the study design does not allow to answer this important question, we are cautious about speculating on the biological reasons behind the lack of association. Similarly, while our data suggest a

potential relationship between *Methanobrevibacter* abundance and treatment success or microbiome health, these observations require further investigation to establish their biological relevance. As noted in the discussion/conclusion, additional studies are necessary to explore these findings in greater detail and to elucidate the underlying mechanisms.

Minor Concerns

Line 44 add word "diagnosed" before "worldwide".

The word “diagnosed” has been added before “worldwide”.

Line 56 change "inflammatory" to "inflammation".

“inflammatory” has been changed to “inflammation”.

Line 65 add "have been approved" to the end of the sentence.

“have been approved” has been added to the end of the sentence.

Line 99 change "casually" to "causally".

“casually” has been changed to “causally”.

Line 307 add comma after "observations"

A comma has been added after “observations”.

Line 200 - add quotation marks to Candidatus, and intestini should not be italicized.

We corrected “intestini”, and added quotation marks to the first appearance of Candidatus.

Line 218 - use correct citation.

The citation has been corrected.

The resolution of the figures is not high.

High resolutions of the figures have been adjusted and sent separately to the journal.

Fig. 1B - Y-axis texts were cut off.

Fig. 1B Y-axis texts have been corrected.

Figure 2A - there is a significance asterisk floating in the top of the graph - what does it represent?

The significance asterisk floating on the top of Figure 2A represents statistical differences between HHC, treated, and untreated groups.

Re: Spectrum02183-24R1 (Reevaluation of the Gastrointestinal Methanogenic Archaeome in Multiple Sclerosis and Its Association with Treatment)

Dear Dr. Christine Moissl-Eichinger:

Your manuscript has been accepted, and I am forwarding it to the ASM production staff for publication. Your paper will first be checked to make sure all elements meet the technical requirements. ASM staff will contact you if anything needs to be revised before copyediting and production can begin. Otherwise, you will be notified when your proofs are ready to be viewed.

Sincerely,
Henning Seedorf
Editor
Microbiology Spectrum

Reviewer #1 (Comments for the Author):

Thank you for addressing my concerns.

Reviewer #2 (Comments for the Author):

The authors have addressed my comments. I have no further concerns.

Reviewer #3 (Comments for the Author):

This revision shows good improvements but falls short of fully addressing this reviewer's previous concerns. In addition, new concerns also arise from the authors' responses.

Major Concerns:

1) The authors selected their patient population by excluding 16S data, longitudinal data and geographic cohorts with a small population. There are three issues here: a) 16S data should still be quite informative, especially looking at the hydrogenotrophic (*Methanobrevibacter*) vs methylotrophic (*Methanomassiliicoccales*, *Methanosphaera*) methanogens; b) longitudinal data should be prioritized over single point data, because the former is more robust; c) excluding certain geographical cohort regardless of sample size implies the microbiome trends are sensitive to geography. If true, then combining samples from very different geographic locations for analysis would not be appropriate. If untrue, combining as many samples as possible would make the analysis more robust.

2) Regarding the significance of *Methanomassiliicoccus*, the authors only reanalyzed the data for PMS vs RRMS, but neither treated vs untreated, nor HHC vs MS. More importantly, a new concern also arises from the reanalyzed data. That is the data points appear very different for the same species between the plots with (Fig. 4 in the rebuttal) and without (Fig. 3C in the main text) *Methanomassiliicoccus*.

3) It is understandable that the authors are cautious about speculating on the biological reasons behind their findings. However, a discussion to put their data in the context of current microbiology knowledge is still needed for this microbiology journal. For example, there are clearly some signs that *Methanomassiliicoccus* (and potentially other methylotrophic methanogens) could be differentially involved in MS. It is also well known that the physiology of *Methanomassiliicoccus* is very different from *Methanobrevibacter*, the former use methylated compounds but the latter prefer other substrates. The authors should reevaluate and discuss these both from a statistical and biological perspective. The biology part should also include both the host and the microbes. For example, do MS patients have more methylated compounds (likely toxic to the host) in their blood/gut/stool compared to healthy controls?

Minor Concerns

- 94: "both host and microbiome"

- Still not sure what the asterisk at the top of figure 2A signifies. I guess it shows if any significance exists in the chart, not sure it is necessary.

ASM Spectrum

REVIEWER'S CHECKLIST

Reevaluation of the Gastrointestinal Methanogenic Archaeome in Multiple Sclerosis and Its Association with Treatment by

Pei Yee Woh, Yehao Chen, Christina Kumpitsch, Rokhsareh Mohammadzadeh, Laura Schmidt, Christine Moissl-Eichinger

WRITTEN COMMENTARY

Abstract and Introduction

The Abstract accurately describes the contents of the paper, and I have no recommendations for it. The Introduction gives an overview of the importance of multiple sclerosis, and the various subtypes. Then an overview of the importance of the gut microbiome, with emphasis on archaeal species. The authors then outline why the data analysis was done and why they chose the dataset they did.

Materials and Methods

I had no issues with this section.

Results

I had no issues with this section.

Discussion

I had no issues with this section.